# Analysis of Spatio-Temporal Transcriptome Profiles of Soybean (*Glycine max*) Tissues during Early Seed Development

**DOI:** 10.3390/ijms21207603

**Published:** 2020-10-14

**Authors:** Shuo Sun, Changyu Yi, Jing Ma, Shoudong Wang, Marta Peirats-Llobet, Mathew G. Lewsey, James Whelan, Huixia Shou

**Affiliations:** 1State Key Laboratory of Plant Physiology and Biochemistry, College of Life Sciences, Zhejiang University, Hangzhou, Zhejiang 310058, China; sunshuo2019@zju.edu.cn (S.S.); majing1002@zju.edu.cn (J.M.); 2Australian Research Council Centre of Excellence in Plant Energy Biology, Department of Animal, Plant and Soil Science, School of Life Science, La Trobe University, Bundoora, Victoria 3086, Australia; C.Yi@latrobe.edu.au (C.Y.); M.Peirats-Llobet@latrobe.edu.au (M.P.-L.); 3Key Laboratory of Soybean Molecular Design Breeding, Northeast Institute of Geography and Agroecology, The Innovative Academy of Seed Design, Chinese Academy of Sciences, Changchun 130102, China; wsd@zju.edu.cn; 4Department of Animal, Plant and Soil Science, AgriBio Building, La Trobe University, Bundoora, Victoria 3086, Australia; M.Lewsey@latrobe.edu.au; 5Australian Research Council Research Hub for Medicinal Agriculture, AgriBio Building, La Trobe University, Bundoora, Victoria 3086, Australia

**Keywords:** *Glycine max*, seed development, transcriptome, RNA-sequencing (RNA-seq), tissue specific, transcription factor (TF), orthologue, gene regulatory network (GRN)

## Abstract

Soybean (*Glycine max*) is an important crop providing oil and protein for both human and animal consumption. Knowing which biological processes take place in specific tissues in a temporal manner will enable directed breeding or synthetic approaches to improve seed quantity and quality. We analyzed a genome-wide transcriptome dataset from embryo, endosperm, endothelium, epidermis, hilum, outer and inner integument and suspensor at the global, heart and cotyledon stages of soybean seed development. The tissue specificity of gene expression was greater than stage specificity, and only three genes were differentially expressed in all seed tissues. Tissues had both unique and shared enriched functional categories of tissue-specifically expressed genes associated with them. Strong spatio-temporal correlation in gene expression was identified using weighted gene co-expression network analysis, with the most co-expression occurring in one seed tissue. Transcription factors with distinct spatiotemporal gene expression programs in each seed tissue were identified as candidate regulators of expression within those tissues. Gene ontology (GO) enrichment of orthogroup clusters revealed the conserved functions and unique roles of orthogroups with similar and contrasting expression patterns in transcript abundance between soybean and Arabidopsis during embryo proper and endosperm development. Key regulators in each seed tissue and hub genes connecting those networks were characterized by constructing gene regulatory networks. Our findings provide an important resource for describing the structure and function of individual soybean seed compartments during early seed development.

## 1. Introduction

Soybean [*Glycine max* (L.) Merr.] is one of the most economically important cultivated plants. Soybean seeds are an excellent source of vegetable oil and protein as well as a significant source of human nutrition and animal feed (Soystats 2020 http://soystats.com). Soybean belongs to the Fabaceae family, one of the largest and most diverse families of flowering plants. Legumes, as this family is known colloquially, have been used extensively as model species in plant biology and historically, soybean and other close relatives have been used as model species to study seed and embryo development [1,2,3,4,5,6]. Plant seed development comprises a series of morphological, physiological, and biochemical changes and can be divided into three major phases: embryogenesis, including cell division and expansion, seed maturation, and desiccation. Seed development is triggered by the double fertilization of the egg cell and central cells that leads to the differentiation of the seed tissues, embryo, endosperm and seed coat [5,7]. These three compartments have different origins and distinct roles in seed formation; the embryo, which is diploid and is derived from one paternal and one maternal genome equivalent; the endosperm, a structure that provides nourishment for the developing embryo, is triploid and derives from one paternal and two maternal genome equivalent; and the seed coat, which provides protection to the embryo and endosperm, has a strictly maternal origin and is derived from the ovule integuments. As a result of the developmental process, the mature seed will consist of a number of different gene expression programs occurring at the same time in the different compartments (e.g., embryo, endosperm, seed coat) as well as within specific regions and tissues (e.g. embryo proper, suspensor, epidermis) [6].

While the genome assembly of soybean (variety Williams 82) was completed in 2010 [8], regulatory networks of gene expression during soybean seed development remain poorly understood. One of the first studies in soybean seed development used RNA-seq to develop an atlas of soybean during seed development [9]. Subsequent studies identified and annotated several seed-specific genes in early seed imbibition [10], as well as analyzing gene expression patterns at all seed developmental stages from post-fertilization to mature seed [11]. The focus of several studies has been on identifying genes controlling agronomically important seed traits such as size, weight, oil and protein contents [12,13,14,15]. For those genes related to seed development, the gene networks and developmental processes controlled by LEAFY COTYLEDON1 (LEC1) were identified by comparing the mRNA profiles of wild-type and lec1-null mutant seeds at several developmental stages [16]. Nonetheless, our understanding of how cellular processes in tissues and regions of tissues are integrated to achieve the coordinated development of soybean seeds is still poor.

Studies of gene expression in seeds of other species have been a useful tool to better understand the biological processes required for proper seed development. For example, such studies have been used to determine that embryo and endosperm have distinct roles during the early development of rice seeds. In the endosperm, expression of plant hormone, galactose metabolism, ribosome and carbon fixation related genes increased significantly from 3 to 6 days after pollination (DAP), whereas genes for defense against disease or response to stress as well as genes for starch/sucrose metabolism were strongly expressed at 10 DAP [17]. In the embryo, metabolism, transcriptional regulation, nucleic acid replication/processing, and signal transduction related genes were expressed predominantly at 3–7 DAP, whilst genes for starch/sucrose metabolism and protein modification were highly expressed at 7–14 DAP [18]. Similarly, a genome-wide analysis of gene expression in several tissues and tissue subregions of Arabidopsis seeds during development revealed several limitations in our knowledge [19]. For example, expression of flavonoid biosynthetic genes was found to be specific to the Arabidopsis seed coat, which may protect seeds against biotic and abiotic stresses, and that the key enzyme required for the synthesis of trehalose is located in chalazal seed coat. The developmentally unique chalazal endosperm expressed the largest number of seed-specific mRNAs. It was concluded this region may serve as a communication hub that integrates seed developmental processes, based upon the finding that it expressed enzymes for the biosynthesis of several hormones, including gibberellic acid, abscisic acid, and cytokinin. A similar study was performed in *Brassica napus,* demonstrating that transcriptional profiles and functional enrichment during the globular stage of *B. napus* seed development is similar to those in the Arabidopsis embryo proper. However, different species have their distinct features during seed development. For instance, the micropylar endosperm in *B. napus* globular seeds was found to significantly accumulate transcripts associated with lipid storage, transport and seed oil body biogenesis during the globular stage, while these GO terms were significantly enriched during maturation rather than in morphogenesis in Arabidopsis. This indicated that *B. napus* may accumulate lipids much earlier than in Arabidopsis [20]. In addition to seed development, the germinating seed is also an excellent model to study genome regulation between cell-types. Recently laser-capture microdissection RNA-seq was applied to plumule, radicle tip and scutellum of germinating barley from which tissue-specific gene expression and storage of transcripts was characterized [21]. In maize, transcriptome analysis of embryo/endosperm interfaces revealed how the embryo and endosperm communicate in order to regulate the production the impermeable cuticle. The endosperm zone adjacent to scutellum was found to have a distinct transcriptional signature, which is specialized in nutrient transport and influenced by the presence of the neighboring growing embryo [22]. In contrast, uncovering this type of comprehensive analysis in soybean is also urgently needed.

Here, we analyzed changes in the transcriptome of tissues of soybean seeds from the globular stage of embryogenesis in early seed development to the cotyledon stage of mature seeds using public RNA-seq data from Gene Networks in Seed Development project (http://seedgenenetwork.net/) [16]. Tissue specific processes, conserved between species, were identified using GO analysis, weighted gene co-expression network analysis (WGCNA) and orthology comparison to and Arabidopsis seed developmental series. These findings enabled the prediction of a set of high-confidence regulators of spatiotemporal-specific gene expression. The resulting analysis provides a comprehensive overview of gene regulation and expression with high spatial and temporal resolution, providing researchers with a roadmap to modify seed characteristics in a tissue specific manner.

## 2. Results and Discussion

### 2.1. Source of Spatial and Temporal Transcriptome Data during Soybean Seed Early Development

We obtained RNA-seq data of soybean seeds (Williams 82) during early seed development from the Gene Networks in Seed Development project (http://seedgenenetwork.net/) [16]. These data were obtained from eight seed compartments (embryo proper, endosperm, endothelium, epidermis, hilum, inner integument, outer integument and suspensor) over three stages of early seed development (globular stage, heart stage and cotyledon stage) using laser capture microdissection (LCM) (Figure 1A). Three biological replicates for each tissue were provided by the resource, comprising approximately 21 million raw reads per library. We mapped raw reads to the soybean reference genome resulting in 79.6 to 85.2% reads successfully aligning (Appendix A). We classified a gene as expressed if its TPM (transcripts per million reads) value was ≥ 5. In total, 40,739 genes, including 2750 transcription factors (TFs), were expressed in at least one of the 72 samples. For each sample, an average of ~15,397 genes and 800 TFs were detected (Appendix A).

To gain further insight into the transcriptome dynamics of early soybean seed development, all the seed tissues and stages (72 samples) were combined and compared using a principal component analysis (PCA) (Figure 1B). The three biological replicates for each sample clustered together, indicating high reproducibility of the data. Overall, the transcriptomes were more similar within the same tissue across the different stages rather than seed tissues clustering within the same stage, similar to gene expression in different tissues of germinating barley seeds [20]. The transcriptome profile in endosperm at the globular stage was distinct from those at the heart and cotyledon stages, which reflects the large changes in endosperm biology with growth occurring at the globular stage but degradation occurring during heart and cotyledon stage. The outer integument transcriptomes also showed differences from heart to cotyledon stage, indicating a change in gene expression patterns in late morphogenesis. In summary, the PCA analysis showed that gene expression in the embryo proper, endosperm and suspensor were more similar, probably because of their maternal and paternal origin, and more distant from the subregions of the seed coat that derived from maternal tissue only.

We analyzed developmental progression in gene expression by comparing the globular stage datasets to the heart and cotyledon stage datasets, respectively. A total of 25,433 genes were differentially expressed (DEGs; FDR < 0.05; Dataset S1). The difference in gene expression was greatest in the endosperm, with 10,242 DEGs between globular and heart stages (S1) and 8999 DEGs between globular and cotyledon stages (S2) (Figure 2A). All seed tissues, except endosperm, had more DEGs in S2, indicating that gene expression in most regions changes dramatically during late morphogenesis when cell differentiation occurs. Nearly half (12,263 of 25,433) of the DEGs were differentially expressed in only one seed tissue, while the other half were differentially expressed over several tissues (Figure 2B). This is similar to previous findings from the seed development of Arabidopsis and barley [18,20]. Only three genes, Glyma.06G012600 (glycine-rich RNA-binding protein 2), Glyma.03G249000 (growth regulating factor-interacting factor1-like) and Glyma.04G122900 (Acyl-CoA-binding protein), were differentially expressed in all seed tissues (Figure 2B). Interestingly, the Arabidopsis glycine-rich RNA-binding protein 2 (soybean homolog Glyma.06G012600) has a role in RNA transcription and processing during stress and the rice homolog exhibited RNA chaperone activity during cold and freezing adaptation to stress [23]. The growth regulating factor-interacting factors (GIF), AtGIF1/ANGUSTIFOLIA3, has a central role in the control of cell proliferation [24] and associates with the core SWI/SNF chromatin-remodeling complex to tightly regulate the transition between cell division and cell expansion in growing leaves [25]. Acyl-CoA-binding proteins (ACBP) had been found to participate in lipid trafficking and signaling. In Arabidopsis, the endoplasmic reticulum-localized AtACBP1 together with other proteins, regulated the generation of sterol signals, which are the putative ligands for TFs required for organ patterning and developmental gene expression [26]. AtACBP1 is localized at the plasma membrane of heart, torpedo, and cotyledon stage embryonic cells demonstrated its role in embryogenesis [27,28,29], while its homolog, AtACBP2, also accumulated in developing embryos at various stages [30].

### 2.2. Tissue-Specific Regulation of Gene Expression Occurs over Developmental Stages

To further understand the dynamics of the global transcriptome during early seed development, we analyzed the expression pattern of all tissue-specific DEGs relative to globular stage. The vast majority of DEGs followed a consistent trajectory of either up-regulation or down-regulation throughout all the stages (Figure 3A). The tissues with highest number of DEGs were endosperm, suspensor and embryo proper, consistent with these tissues being the most biochemically active tissues in the seed. These results highlight extensive tissue-specific gene expression during early seed development, indicating that the expression of the soybean genome is regulated in a tissue-specific manner. The potential functions of the tissue-specific DEGs were investigated in order to gain insight into the biological processes unique to each tissue. To achieve this, we applied PageMan over-representation analysis (z-score, *p*-value < 0.05, Figure 3B, Dataset S2). Significantly over-represented, up-regulated genes in the endosperm were involved in lipid and nucleotide metabolism and chromatin and cytoskeleton organization (Figure 3B). After fertilization in the endosperm, endoreduplication (DNA replication occurs without cytokinesis) occurs and the endosperm cells enlarge using microtubules that emanate from the nuclear envelope and determine the size and shape of the future cells. Microtubule arrangement has been shown to be important in the cellularization of the endosperm [31,32]. The involvement of cytoskeletal elements in protein body formation has also been proposed [33]. Contrastingly, proteolysis was significantly enriched amongst the down-regulated endosperm specific DEGs, which is consistent with the function of protein storage in this tissue [34].

The hilum is the tissue that connects the seed to the pod and gene expression in this tissue is not well studied. Our analysis determined that highly up-regulated hilum-specific genes were involved in vesicle trafficking and solute transport. It is also evident from its role in nutrition, lipid and carbohydrates metabolism and polyamine degradation (Figure 3B). These results indicated the importance of this tissue in nutrition and pod-seed communication. Contrastingly, for a sub-region of seed coat, the integuments, tissue-specific regulation of genes involved in nutrient uptake (nitrogen and phosphorus), carbohydrate and lipid metabolism was evident. These results reflect the importance of nutrient allocation during early seed development. Our analysis demonstrated a special role of outer integument in controlling of cytoskeleton organization, specially, nuclear dynamics and plastid movement, which were significantly enriched amongst the down-regulated outer integument-specific DEGs. 

### 2.3. Predicting Candidate Transcriptional Regulators Using Weighted Gene Correlation Network Analysis

We defined co-expressed cohorts of genes (termed modules) in order to identify potential spatiotemporal-specific regulators that may underpin the tissue specificity of the gene expression outlined above. To do so we applied weighted gene correlation network analysis (WGCNA). Genes were filtered prior to WGCNA to remove those with low variability (Coefficient of Variance < 0.5) or expression (TPM < 5) across samples because these would not contribute to the network, leaving 30,930 genes for analysis. After generating a summary profile (eigengene) for each module, we observed that 18 of the 21 modules were predominantly co-expressed in a single seed tissue (Figure 4). Three out of 18 co-expression modules, including the brown, grey60 and purple modules, were specifically expressed in embryo proper. Notably, the brown module, which contains 1289 genes, was mainly enriched in protein translation and ribosome biogenesis related processes. Ribosomal proteins were highly expressed in the globular stage and coordinated with embryogenesis in Arabidopsis [35]. Therefore, the brown module genes might be responsible for the initiation of embryogenesis. Moreover, the gradual reduction of expression of brown module genes from globular to cotyledon stage implies the ribosome biogenesis activity was gradually decreasing during embryo proper establishment, with the ribosome machinery established for germination where translation is essential and a very early process [36]. It is known that photosynthesis occurs in the embryo, and embryonic photosynthesis affects post-germination plant growth [18,35,37,38]. In line with this, grey60 and purple modules were significantly enriched in photosynthesis related GO terms, indicating embryonic photosynthesis might occur during the heart stage and cotyledon stage. It is notable that, unlike the brown module, the expression of purple module genes increased from globular to cotyledon stage, suggesting a functional transition of the embryo from globular to cotyledon state (Figure 4B, Dataset S3). The integuments, including outer and inner integuments, are the interface of maternal tissues and endosperm. Interestingly, the genes from integument modules, such as light green, turquoise and greenyellow, were significantly enriched in transport related functions including transmembrane transport, protein transport and carbohydrate transport. These enriched functions suggest the integuments may play an important role in transporting metabolites such as protein and carbohydrates from maternal tissues to the endosperm and embryo for their growth. The epidermal cells of the seed coat protect seeds against environmental stresses, partly by accumulating toxic compounds such as flavonoids and terpenoids [39,40]. In agreement with this, the epidermis-specific yellow module was highly enriched in flavonoid biosynthesis processes, confirming the protective function of the seed coat. The suspensor is a supportive structure that pushes the embryo close to the nourishing endosperm and is involved in nutrient and hormone regulation, such as auxin transport [18,41,42]. Consistent with this, auxin-activated signaling pathway genes were enriched in the pink module, which was predominantly expressed in the suspensor. Two genes (Glyma.02G007300, Glyma.10G138500), which were both annotated as indole-3-acetic acid inducible 30 (IAA30), were strongly associated with the pink module (kMEs 0.99 and 0.98, respectively).

We identified 332 transcriptional regulators (Appendix A) that were candidate regulators of the modules using the following criteria; (a) TFs having high correlation with the expression profile of the module to which they belonged (kME > 0.9, termed hub genes), and (b) TFs themselves significantly differentially expressed (FDR < 0.05, Dataset S1). Twenty of the 21 modules contained candidates, which encompassed a wide range of transcriptional regulator families. Most TFs from the same family were predicted to co-regulate modules from one or two tissues. For instance, all 15 plant-specific transcription factor YABBY family members were only identified as hub regulators in embryo proper modules. The YABBY family TFs were only found in seed plants and are involved in the regulation of adaxial–abaxial polarity and development of lateral organs [43,44,45,46]. The enrichment of YABBY family genes in the embryo properly indicated that these TFs might also have an important role in embryo development. Interestingly, there were six TFs belonging to four different families specifically expressed in the black, endosperm-specific module, including ETHYLENE RESPONSE FACTOR 1 (ERF1, Glyma.13G122700), CYTOKININ RESPONSE FACTOR 4 (Glyma.14G123900), MADS-box transcription factor family protein (Glyma.05G140000 and Glyma.18G053400) and TELOMERASE ACTIVATOR 1 (TAC1, Glyma.10G184500 and Glyma.20G206000). Genetic variation of *ERF1* was associated with thermotolerance in lettuce seed germination [47]. Our results imply that ERF1 might not only play a role in seed germination, but also seed development, specifically, endosperm development. Telomerase expression is mainly observed in reproductive organs and rapidly dividing cells [48,49]. After fertilization, endosperm undergoes rapid proliferation without the synthesis of cytoplasm, cell membranes and cell walls, and cell division in endosperm is faster than in embryo [50,51]. Given the high expression of TAC1 in endosperm, it is reasonable to think that TAC1 is critical during the endosperm development. However, there were some transcription factor families predicted to regulate more than one seed tissue, such as AP2, bHLH, BLH, C2H2, homeodomain, MYB, NAC and WRKY. For example, one soybean homologue of AtNAC87 in Arabidopsis (Glyma.19G109100), which has been reported to be involved in seed development [52], was identified as a candidate regulator of the inner integument. However, several other NAC family genes, such as NAC2, NAC44 and NAC47, were mainly expressed in endosperm, suspensor and endothelium (Appendix A). The Arabidopsis *AGAMOUS* (AG) encodes a MADS box transcription factor, which is involved in flower development [53,54]. Several homologues of AG in *Brassica napus* and strawberry (*Fragaria vesca*) were predicted as putative regulators of seed coat development [19,55]. We observed 15 AGAMOUS-like (AGL) genes that were highly expressed in the endosperm and the endothelium, indicating AGL genes might also regulate endosperm and endothelium development in soybean. The identification of many distinct modules of co-expressed genes with distinct expression profiles that contain a large number of candidate regulators, illustrates the complexity of the tissue-specific regulation in the soybean seed.

### 2.4. Integration of Orthology and Expression Information Identifies Conserved Orthologues between Soybean and Arabidopsis

Soybean and *Arabidopsis thaliana* are two important model organisms for dicots, both of which progress through globular, heart and cotyledon stages during early seed development. To carry out an unbiased comparison of seed development transcriptomes between these species, public Arabidopsis seed development datasets were obtained from Gene Expression Omnibus (GEO) database (GSE11262, GSE15160, and GSE12403) [19,56]. OrthoFinder was used to define sets of orthologous genes in the two species [57]. A total of 12,344 orthogroups were defined between soybean and Arabidopsis representing 18,110 genes in soybean and 26,975 genes in Arabidopsis (Appendix A). Orthogroups containing genes in soybean and Arabidopsis were subsequently analyzed with Clust to define sets of co-expressed orthologous genes within and across species [58]. The embryo proper and micropylar endosperm data were available in the public Arabidopsis data; thus, we only compared these two tissues here.

For soybean and Arabidopsis embryo proper, a total of 2,355 and 4,131 orthogroups were distributed in seven clusters, respectively (Figure 5, Appendix A). The gene expression pattern in clusters 1, 2 and 3 was highly conserved between soybean and Arabidopsis. Subsequently, a GO term enrichment analysis was performed for the genes in the orthogroups included in each of the clusters (Appendix A). Expression of genes in cluster 1, which represents 44% and 35% of the orthogroups in soybean and Arabidopsis, respectively, declined during seed early development. Genes in this cluster were associated with enriched GO terms related to RNA processing, mRNA processing and embryo sac egg cell differentiation. These likely represented the breakdown of RNA with modified functions reserved in embryo proper during seed development. By contrast, the orthogroups in cluster 3 contained genes for which transcript abundance increased and the GO term enrichment was associated with biosynthetic cysteine and fatty acid biosynthetic processes. These indicated the requirements for the synthesis of amino acids during seed development in soybean and Arabidopsis embryo proper. Another conserved cluster, cluster 2, included those orthogroups whose expression peaked at heart stage. The genes of orthogroups in this cluster were associated with DNA recombination in soybean and oxidation-reduction and steroid metabolism in Arabidopsis. Other clusters (4, 5, 6 and 7) contained genes with opposing co-expression profiles between the two species. This highlights that, whilst processes may be conserved overall, the spatiotemporal expression of some genes will have diverged between species, and consequently translational comparative studies from one species to another are essential.

### 2.5. Tissue-Specific Expression of Transcriptional Regulators during Early Seed Development

To understand how tissue-specific genome regulatory programs are established we analyzed the differential expression of TFs between soybean tissues. In total, 879 tissue-specific TFs were identified (Dataset S4). We reasoned that these were strong candidates to control tissue-specific genome regulatory programs and wanted to determine which families of TFs were operating in each seed tissue. A total of nine out of ten families were significantly enriched amongst the DEGs specific to one seed tissue (Figure 6A; *P* < 0.005; embryo proper - YABBY, TALE; endosperm - M-type_MADS, MIKC_MADS, B3, BBR-BPC; epidermis - EIL; hilum - MYB; suspensor - NAC). The ERF TF family was the only enriched family shared between tissues (endothelium and epidermis). There was no enrichment of any transcriptional regulator family in inner or outer integument specific DEGs.

Binding of TFs to regulatory regions of the genome usually requires the presence of DNA motifs that TFs recognize. We examined DNA motifs enriched in the promoter regions of our tissue-specific DEGs in order to identify the families of TFs likely to be active in each seed tissue (Figure 6B). Several motifs significantly associated with promoters of DEGs were identified de novo in each tissue (E-value < 1e-10). The BASIC PENTACYSTEINE (BPC) family (BPC1, BPC5 and BPC6) had the most enriched motifs in the promoters of DEGs in embryo proper, endosperm, endothelium, epidermis and hilum. The BPC TF family is integral in a wide range of processes that support plant normal growth and development [59]. Another universally enriched motif, associated with TF OBP3, was identified in embryo proper, endosperm and suspensor, but not significantly enriched in any sub-region of seed coat. Mis-expression of this TF correlates with growth defects [60]. Our results also supported the tissue-specific regulation of OBP3. Similarly, RAMOSA1, NUC and MGP were only significantly enriched in embryo proper and endosperm. In contrast, Adof1 was only enriched in the sub-regions of seed coat (hilum and outer integument). The most enriched motif in hilum had no significant similarity to motifs from previously characterized plant TFs, demonstrating that there are uncharacterized transcriptional regulators active in hilum. These analyses considered together indicated that a diverse range of TFs were likely to contribute to genome regulation and have tissue-specific and common features during early seed development. Our analysis provided a better understanding of the spatiotemporal expression of known transcriptional regulators.

### 2.6. Gene Regulatory Network (GRN) of the Seed Tissue-Specific Genes

GRNs are a useful tool to better understand the relationships between regulators and their targets. In this study, we inferred the GRN for each seed tissue separately (eight networks for each tissue) using genist algorithm from TUXNET [61] and then combined them to generate the final network. To discover the tissue-specific regulatory relationship among the key regulators for each network, genes from two sets were used for network inference. The first gene set included genes that overlapped between tissue-specific DEGs and hub genes (kME > 0.9) of this tissue specific WGCNA module, and the second set contained genes (ubiquitous genes) that were differentially expressed between multiple tissues (including this tissue) and also identified as hub genes in the non-tissue-specific WGCNA modules (red, grey, and lightcyan). In order to obtain robust final networks, only the most reliable regulatory relationship was retained. Interestingly, almost all genes in each tissue were regulated by one or two genes, which helped us find the core genes (Figure 7, Appendix A).

The endosperm network was the largest, with 600 interactions including 259 endosperm-specific and 43 ubiquitous genes. Interestingly, most endosperm-specific genes (205/258) in this network, including a set of AGAMOUS-LIKE (AGL) genes (AGL6, AGL20, AGL61, AGL62, AGL80), were activated by an endosperm-specific gene AGL96 (Figure 7A). Although the function of AGL96 is still unknown, six AGLs (AGL36, AGL37, AGL38, AGL40, AGL62, AGL90) were co-expressed in the endosperm and absent in the embryo in Arabidopsis [62]. As the reported central hub of the AGL cluster in Arabidopsis, AGL62 was revealed to be essential for the normal development of seeds and avoid premature death [63]. Although most members of Arabidopsis AGL co-expression cluster were not consistent with the endosperm-specific network in soybean, AGL96, the ortholog of the core regulator of the Arabidopsis AGL co-expression cluster AGL62, was identified in a core position in the soybean network. This indicated that similar AGL clusters may also play a critical role in maintaining seed activity during soybean endosperm development.

Unlike the AGL genes that mainly functioned in the endosperm, three SCARECROW-LIKE (SCL) genes were found as ubiquitous genes. For example, two paralogue genes, SCL28a and SCL28b, regulated all genes in soybean suspensor and endosperm networks, respectively. In Arabidopsis root, SCL28 was found to orchestrate cell division and elongation under various environmental conditions [64]. This suggests SCL28 may promote soybean seed growth by regulating cell division in the suspensor and endosperm. In addition, another SCL ubiquitous gene, SCL3, which is involved in gibberellin signaling by antagonizing master growth repressor DELLA in Arabidopsis seeds [65], was predicted to regulate nearly all ubiquitous genes in epidermis and endothelium. Given SCL3 was predicted to be repressed by SCL28a, SCL28 may promote soybean seed development by indirectly regulating gibberellin homeostasis.

In the hilum-specific network, all genes were inferred to be regulated by AUXIN RESPONSE FACTOR16 (ARF16) and TINY2 (Figure 7A). This was consistent with their roles of promoting and inhibiting cell growth and proliferation by responding to auxin and brassinosteroid in Arabidopsis [66,67]. Moreover, a ubiquitous gene E2F1, which was reported to regulate cell proliferation by acting as a target of auxin in Arabidopsis [68], was predicted to be repressed by ARF16 and TINY2 in soybean (Figure 7B). In addition, E2F1 was also inferred to be negatively regulated by SCL3, whose potential involvement in gibberellin signaling in epidermis and endothelium has been described in the previous paragraph. These indicated that in the soybean seed coat the above core regulators may form a previously uncharacterized pathway, coordinating with plant hormones to regulate seed development. In summary, our analysis provided a solid foundation for the identification of key regulators active in seed tissues during early seed development.

## 3. Summary

In this study, we analyzed a transcriptomic dataset from 8 seed tissues (embryo proper, endosperm, suspensor, endothelium, epidermis, hilum, inner integument and outer integument) at 3 developmental stages (globular stage, heart stage and cotyledon stage) during soybean seed morphogenesis (Figure 1). Our analyses determined that tissue specificity of gene expression was stronger than developmental stage specificity, as reported in other plant species [19,21]. Nearly half of the DEGs detected were specific to an individual seed tissue.

Many of the predicted gene expression regulators identified by our GRN functioned tissue-specifically, which is consistent with the large proportion of tissue-specific gene expression during seed development. However, not all regulators were predicted to have tissue-specific function. This likely indicates that establishing the unique functions of individual tissues requires the activity of regulators specific to each tissue, but that there is also a set of core cellular functions conserved between a number of tissues. Many predicted soybean regulators had Arabidopsis orthologs with known functions in seed development. The GRN approach predicted a previously uncharacterized role for the transcription factor AGL96, proposing that it is a central regulator of endosperm gene expression. This may be comparable to its ortholog AGL62, which is co-expressed with a group of AGL transcription factors in the Arabidopsis endosperm but not expressed in the embryo of that species. Overall, the GRN analyses indicated that there is relatively high connectivity between tissue-specific and ubiquitous regulators during seed development, including repression, activation and feedback loops.

A large proportion of the genes differentially expressed in the embryo proper and endosperm during seed development was also conserved between soybean and Arabidopsis. Using a strict definition of orthology, ~65% of orthologous genes exhibited the same trend in transcript abundance between species in the embryo proper (Clusters 1, 2 and 3, Figure 5). However, the remaining ~35% exhibited opposite trends in the expression between species. The differing annotation quality of the soybean and Arabidopsis genomes makes direct comparison of functional enrichment between species challenging. However, based on the clear conservation in gene expression trends alone, we can conclude that there are likely conserved units comprised of orthologous genes that perform base seed developmental processes. Contrastingly, the cases of opposite expression trends likely indicate processes that vary between the seeds of the two species, due to their different morphology and biochemistry. These differences are important to consider when designing translational applications of GRNs that use data from another species.

Our study as a whole provides a detailed spatiotemporal view of gene expression and regulation during early soybean seed development. Our work provides the research community with a better understanding of the expression characteristics and functions of different seed compartments. Insight into the regulators and their targets during seed development is an essential tool when designing strategies for the biotechnological improvement of seed performance.

## 4. Materials and Methods

### 4.1. RNA-Seq Data Quality Control Analysis

FASTQC (http://www.bioinformatics.babraham.ac.uk/projects/fastqc/) was used for initial reads quality control (QC) metrics (base quality distribution). NGS QC toolkits were used to trim adaptors and low-quality reads [69]. The *G. max* reference genome (Wm82.a4.v1) and annotation files were downloaded from Phytozome V13 (https://phytozome.jgi.doe.gov) [70]. RNA-seq reads were mapped to the *G. max* reference genome using hisat2 V2.1.0 with default settings [71]. The bam files of uniquely mapped reads were used as inputs for the Stringtie [72], and TPM (transcripts per million reads) values were calculated to measure the expression levels of genes. To reduce the data noise, we defined a gene as expressed if its TPM was ≥ 5. To determine the variation of the RNA-seq samples, we performed principle component analysis (PCA) using plotPCA function in DESeq2 R package [73].

### 4.2. PageMan Analysis of DEGs

Lowly expressed genes (average TPM < 5) were filtered out and differential expression analyses was carried out using edgeR [74]. DEGs were identified as those with a *P* < 0.05 and FDR < 0.05. UpsetR was used to show overlapping and specific responses in tissues and stages [75]. The significant DEGs were analyzed for overrepresented functional categories using PageMan (Mapman v. 3.5.1 R2) [76]. To do this, a custom mapping file for soybean was generated using Mercator, with all available query databases. Using this mapping file and the DEGs, over-represented functional categories were identified among the DEG sets using Fisher’s test (*P* < 0.05). Given the highly conserved changes in differential expression over stages, a single DEG set was generated for each tissue using the maximum (for up-regulated genes) or minimum (for down-regulated genes) fold-changes. The outputs from these PageMan analysis were then visualized as heatmaps indicating the z-scores (> 1.96 indicates *P* < 0.05).

### 4.3. Identification of Co-Expression Modules and Hub Genes

Prior to analyze co-expressed genes, we calculated the coefficient of variation (CV) of each gene based on their TPM values and filtered the genes with low CVs (< 0.5), removing low expression and/or low variability of expression across samples. Next, the R package WGCNA (Weighted Correlation Network Analysis) was used to identify modules of highly correlated genes across all samples [77]. The parameters of WGCNA used default settings, except for the power was 12, network type was signed, minModuleSize was 30. Eigengenes and clusters were calculated based on the correlations to quantify the co-expression similarity of entire modules, using a strict cut-off of 0.25, corresponding to correlation of 0.75. The eigengene expression values were calculated for each sample in each module. Then, kME, known as the module membership value, was calculated using SignedKME algorithm to represent the correlation between a gene and the module eigengene value. Intramodular hub genes were identified if their kME > 0.9.

### 4.4. Orthologues Identification between Soybean and Arabidopsis

Orthologues and corresponding orthogroups between Arabidopsis and soybean were inferred via OrthoFinder v. 2.3.12 (https://github.com/davidemms/OrthoFinder) with default parameters and MMseqs2 for sequence similarity searches [78]. Protein sequences used here correspond to the longest transcripts of all genes of soybean (Wm82.a4.v1 release) and Arabidopsis (TAIR10 release), and they were acquired from phytozome v13 (https://phytozome-next.jgi.doe.gov/). The expression data of Arabidopsis seed were acquired from the Gene Expression Omnibus (GEO) database (accession no. GSE11262, GSE15160, GSE12403). Co-expression analysis of orthogroups was performed using the Clust software v. 1.12.0 with default parameters [58].

### 4.5. Transcription Factor and Promoter Motif Enrichment

All soybean TFs were annotated using PlantTFDB v. 5.0 (http://planttfdb.cbi.pku.edu.cn) [79]. Significantly enriched TF families were identified by R package clusterProfiler (*p*-value < 0.005) [80]. All TFs expressed in this study were defined as background genes, those tissue-specific DE-TFs were used for enrichment. The promoter regions (1000 bases upstream the transcription start site) of tissue-specific DEGs were examined to identify enriched DNA motifs that may correspond to TF-binding sites. These were analyzed using MEME-ChIP (http://meme-suite.org/tools/meme-chip) with default parameters against the built-in motif library and a background of all soybean promoter regions [81]. Enriched motifs were reported if E-value < 1e-10, as suggested by the MEME-ChIP documentation. Significant similarities to known plant motif families were reported. Similarity to non-plant motifs was not reported.

### 4.6. Gene Regulatory Network Inference

Networks were inferred for each seed tissue separately (resulting in eight networks, one for each tissue) using genist algorithm from TUXNET [61] and then combined to form the final network. For each network, only a) tissue-specific DEGs and overlapped with hub genes of tissue-specific WGCNA modules (kME > 0.9) and b) DEG differentially expressed in multiple tissues (ubiquitous) and overlapped with hub genes of non-tissue-specific WGCNA modules were used in the network inference. To get the regulatory relationship of the genes both in the same time point or in the following time point, the time lapse value was set as (0,1). Other parameters of GENIST were set with default values.

## Figures and Tables

**Figure 1 ijms-21-07603-f001:**
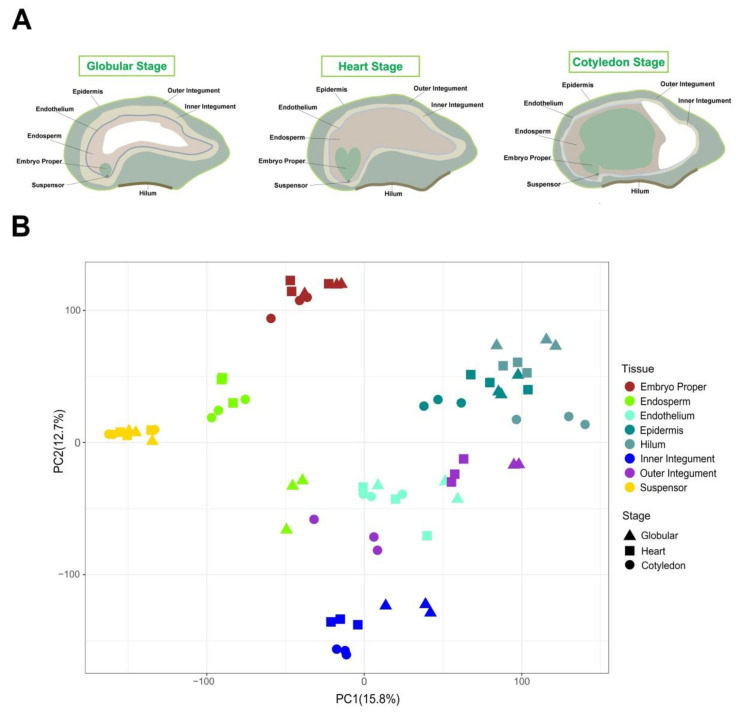
Transcriptome relationships amongst the 72 samples of early soybean seed development from the Gene Networks in Seed Development project. (**A**) Schematic diagram of the soybean seed over three developmental stages showing the various tissues dissected for analysis. (**B**) Principal Component Analysis (PCA) of the transcriptomes of three development stages and eight seed tissues.

**Figure 2 ijms-21-07603-f002:**
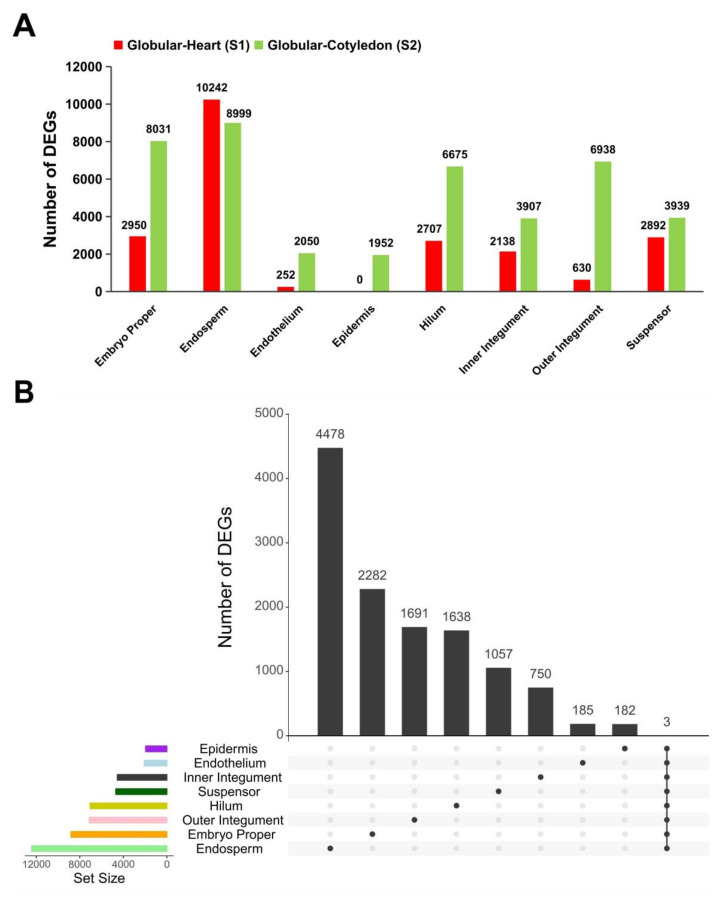
Overlapping and specific responses in soybean seed tissues. (**A**) Number of DEGs between globular-heart stage, and globular-cotyledon stage. (**B**) UpSetR plot shows overlapping and specific responses in seed tissues.

**Figure 3 ijms-21-07603-f003:**
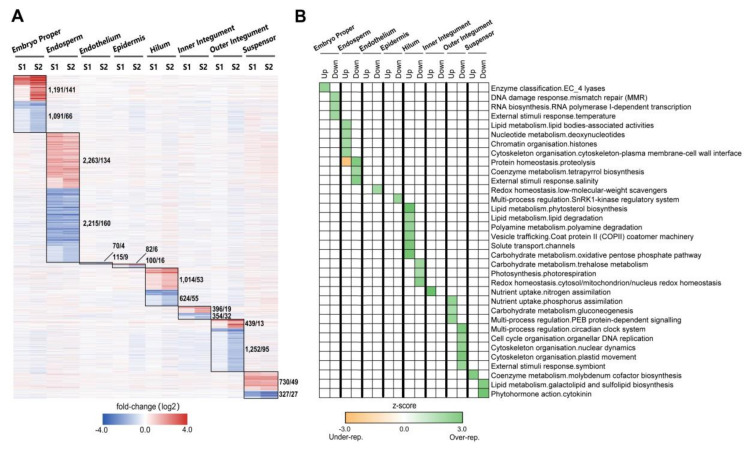
Functional analysis of tissue-specific DEGs. (**A**) Hierarchical clustering of fold-changes in transcript abundance for all specific DEGs in each seed tissue. S1, Heart stage relative to Globular stage; S2, Cotyledon stage relative to Globular stage. The number of genes and TFs in each box are listed on the right. (**B**) PageMan over-representation analysis (Fisher’s test, *P* <0.05) of specific DEGs in each seed tissue. Over-represented and under-represented functional categories specific in one tissue are shown (full output in Dataset S2).

**Figure 4 ijms-21-07603-f004:**
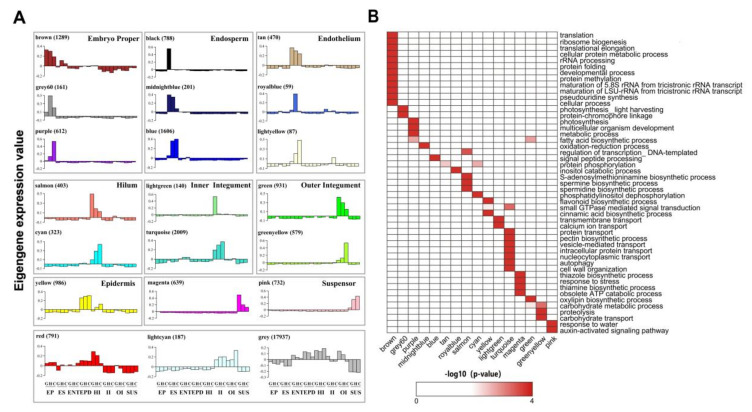
Eigengene expression patterns for each module eigengene. (**A**) Eigengenes represent the expression of their co-expression modules across all samples, in arbitrary units. We categorize the eigengenes according to the seed tissue(s) in which they are predominantly expressed. Numbers in parentheses correspond to the number of member genes in the module. G-globular stage, H-heart stage, C-cotyledon stage, EP-embryo proper, ES-endosperm, ENT-endothelium, EPD-epidermis, HI-hilum, II-inner integument, OI-outer integument, and SUS-suspensor. (**B**) Significant enriched GO terms (biological process) of tissue specific WGCNA co-expression modules (*P* < 0.001).

**Figure 5 ijms-21-07603-f005:**
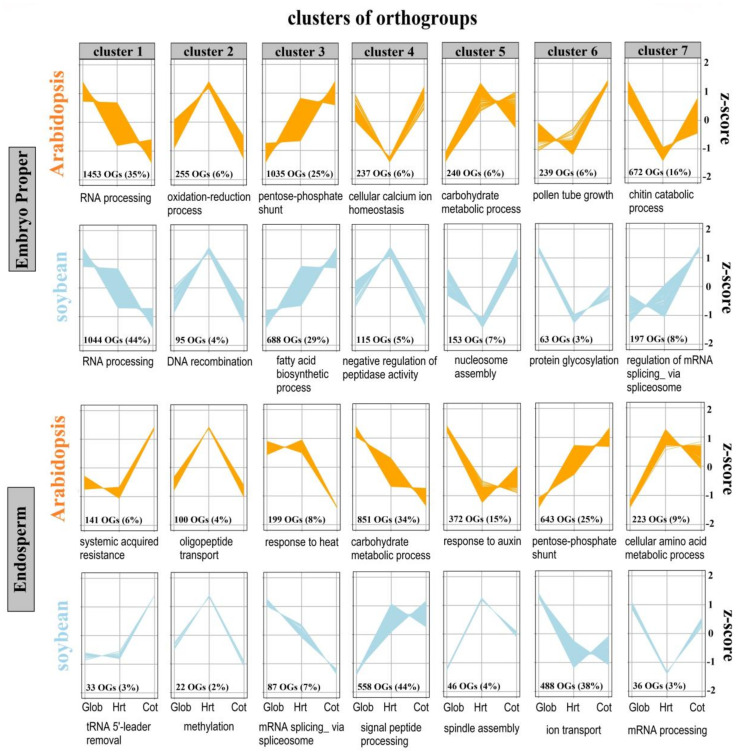
Comparison of embryo proper and endosperm transcript abundance profiles during seed development in soybean and Arabidopsis. For cross-species comparison of seed development, changes in transcript abundance in soybean were compared to published data in Arabidopsis [19]. The numbers in each panel correspond to the number of orthogroups in each cluster and the percentage of all orthogroups per species. The most significantly enriched GO term in each cluster is given below each panel, and the full list is available in Appendix A.

**Figure 6 ijms-21-07603-f006:**
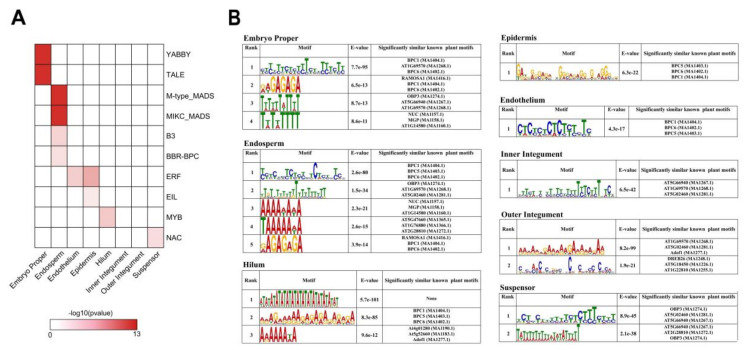
Genes encoding transcriptional regulators are tissue-specifically expressed during soybean seed development. (**A**) Transcription factor family enrichment of tissue-specific differentially expressed transcription factors in each seed tissue (*P* < 0.005). (**B**) Motifs significantly enriched (E≤1e-10) in promoters of genes specific expressed in each seed tissue were identified de novo. Enriched motifs were compared to known motifs and families of significant matches to plant transcription factors are indicated.

**Figure 7 ijms-21-07603-f007:**
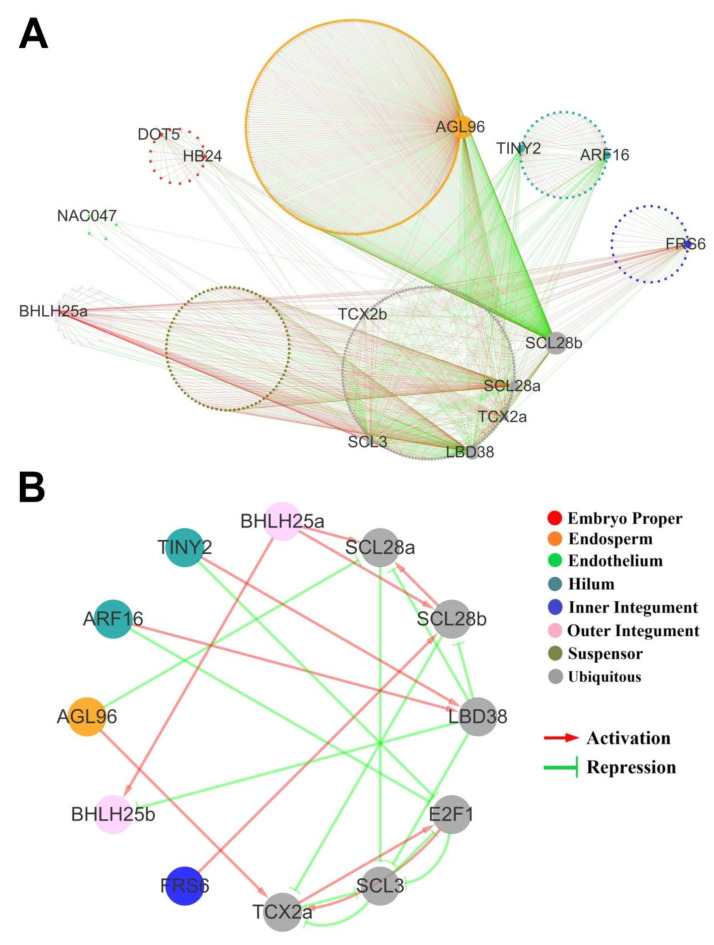
Predicted gene regulatory network (GRN) of early developmental soybean seed. (**A**) Network hubs regulating genes in each seed tissue. Node size represents the degree and key regulators with high degree are highlighted. Node colors represent the tissues which the DEG belong to. Edge colors represent the regulatory relationships is inferred. Red: activation; Green: repression. (**B**) The regulatory relationship between the key regulators in each tissue and ubiquitous regulators with high degree.

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
