# Peer review of "Analysis of Spatio-Temporal Transcriptome Profiles of Soybean (Glycine max) Tissues during Early Seed Development"

_ijms, 2020, doi:10.3390/ijms21207603_

Round 1
Reviewer 1 Report
Dear Authors,
Manuscript titled „Analysis of Spatio-temporal Transcriptome Profiles of Soybean (Glycine max) Tissues during Early Seed Development” is written in an understandable and transparent way. The experiments were well planned and the results have been analysed in detail and thoroughly. The only minor drawback of the manuscript seems to be the division of manuscript into chapters especially chapters Results and Discussion. First of all, the results are already discussed in the chapter results, while the chapter discussion lacks any discussion with other scientific papers, as evidenced by only two citations in the whole chapter discussion and it is more like a summary or conclusions. Therefore, it seems to me that the name of the "results" chapter should be changed to "results and discussion" and the discussion chapter should be called summary. Nevertheless, the presented results are very interesting. Therefore, I recommend to publish manuscript titled „Analysis of Spatio-temporal Transcriptome Profiles of Soybean (Glycine max) Tissues during Early Seed Development” in International Journal of Molecular Sciences after minor revision.
Author Response
Response to Reviewer 1 Comments
Point 1: Manuscript titled ”Analysis of Spatio-temporal Transcriptome Profiles of Soybean (Glycine max) Tissues during Early Seed Development”is written in an understandable and transparent way. The experiments were well planned and the results have been analysed in detail and thoroughly. The only minor drawback of the manuscript seems to be the division of manuscript into chapters especially chapters Results and Discussion. First of all, the results are already discussed in the chapter results, while the chapter discussion lacks any discussion with other scientific papers, as evidenced by only two citations in the whole chapter discussion and it is more like a summary or conclusions. Therefore, it seems to me that the name of the "results" chapter should be changed to "results and discussion" and the discussion chapter should be called summary. Nevertheless, the presented results are very interesting. Therefore, I recommend to publish manuscript titled ”Analysis of Spatio-temporal Transcriptome Profiles of Soybean (Glycine max) Tissues during Early Seed Development” in International Journal of Molecular Sciences after minor revision.
Response 1: Thank you for the suggestion. We have changed the "results" to "results and discussion", and "discussion" to "summary" respectively.
Reviewer 2 Report
The authors have performed a descriptive study of gene expression throughout development. It is a study that integrates a lot of already available data. The analyses done show that the obtained data(sets)/results are reliable. I agree with the authors that this study provides a valuable resource.
Nice introduction. Overall, I find, the text is clearly written and well structured. Sometimes there’s a little discussion in the results, but this aids in understanding and I do not find it a bad thing. The discussion therefore tends to be a bit more summarising here and there. Not a problem, the paper reads well.
Questions:
Line 121-122/460-464: “We defined a gene as expressed if its TPM (transcripts per million reads) value was ≥ 5.”. Wouldn’t shorter genes automatically have less reads than longer genes? Or is this what you mean with transcripts?
I am unfamiliar with this database for soybean. Does it make sense to state which studies were include? It seems that there are about 250 RNAseq studies, did you make a selection? Obviously you used the raw reads, but the methods to isolate RNA, create libraries and the sequencing can influence things? I am aware the PCA plot looks good, but might be good to comment on this.
You used datasets for the Williams 82 ecotype, but used a different reference genome for the mapping, right? TFs might be more conserved, are generally shorter… Do you think this might have any influence? Again, I’m not very familiar with these aspects of soybean.
For the material and methods, to my knowledge, the right checks and controls have been used during the bioinformatic analyses.
Minor comments
Line. 149: a random “the” at the end of the sentence? Didn’t you compare globular and heart, and globular and cotyledon? Review sentence.
Figure 3B: is it possible to make thicker vertical lines between the tissues (the two columns belonging to one tissue more clearly separated)?
Line 188-189: “.. has recently been proposed.” Reference?
Fig. 4. Can you please improve the quality (of the legend for the tissues)?
Fig. 5: enlarge the GO terms given in the figure.
Author Response
Response to Reviewer 2 questions
Point 1: Line 121-122/460-464: “We defined a gene as expressed if its TPM (transcripts per million reads) value was ≥ 5.”. Wouldn’t shorter genes automatically have less reads than longer genes? Or is this what you mean with transcripts?
Response 1: Yes, shorter genes generally have less reads than longer genes, so we used TPM to avoid the influence of transcript length on transcript expression. What we want to do here is just filtering the low expressed genes.
Point 2: I am unfamiliar with this database for soybean. Does it make sense to state which studies were include? It seems that there are about 250 RNAseq studies, did you make a selection? Obviously you used the raw reads, but the methods to isolate RNA, create libraries and the sequencing can influence things? I am aware the PCA plot looks good, but might be good to comment on this.
Response 2: Yes, as you mentioned there are 246 RNAseq datasets in this seed database, but actually not only for soybean, there are also 6 Arabidopsis satasets, 4 Scarlet Runner Bean datasets and 4 Common Bean datasets. In this study, we selected 72 soybean samples from a same study (Pelletier et al., 2017) and the methods to isolate RNA, create libraries and the sequencing were finished at the same time, so we think this wouldn’t have an influence on these aspects.
Point 3: You used datasets for the Williams 82 ecotype, but used a different reference genome for the mapping, right? TFs might be more conserved, are generally shorter… Do you think this might have any influence? Again, I’m not very familiar with these aspects of soybean.
Response 3: Not yet. The reference genome we used for the mapping is also the Williams 82. So we think this wouldn’t have an influence on this aspect.
Response to Reviewer 2 comments
Point 1: Line. 149: a random “the” at the end of the sentence? Didn’t you compare globular and heart, and globular and cotyledon? Review sentence.
Response 1: This sentence is at line 152-153 now. Sorry for the ambiguity, as you said, we actually compared globular and heart, and globular and cotyledon. To avoid the ambiguity, we changed the sentence to "We analyzed developmental progression in gene expression by comparing the globular stage datasets to the heart and cotyledon stage datasets respectively. "
Point 2: Figure 3B: is it possible to make thicker vertical lines between the tissues (the two columns belonging to one tissue more clearly separated)?
Response 2: Good suggestion. We have added 7 thicker vertical lines in Fig3B to make columns belonging to one tissue more clearly seperated.
Point 3: Line 188-189: “.. has recently been proposed.” Reference?
Response 3: This sentence is at line 194 now. Sorry for not citing the reference, we have added the reference after this sentence. Due to this paper was published in 2015, we removed the adverb "recently". After that, all the reference numbers after this place have also been changed.
Point 4: Fig. 4. Can you please improve the quality (of the legend for the tissues)?
Response 4: Yes, we have tried to make it clear by enlarging the legend, to achieve this goal, we use the abbreviation instead of the full name of each stage and tissue. And we add the explanation in the figure legend: G-globular stage, H-heart stage, C-cotyledon stage, EP-embryo proper, ES-endosperm, ENT-endothelium, EPD-epidermis, HI-hilum, II-inner integument, OI-outer integument, SUS-suspensor.
Point 5: enlarge the GO terms given in the figure.
Response 5: Thanks for the suggestion, we have enlarged the GO terms given in the Fig.5. But to achieve this goal, only the most significant enriched one was listed instead of two or three GO terms.